# Plant-Based Diets and Lipid, Lipoprotein, and Inflammatory Biomarkers of Cardiovascular Disease: A Review of Observational and Interventional Studies

**DOI:** 10.3390/nu14245371

**Published:** 2022-12-17

**Authors:** Patrick S. Elliott, Soraeya S. Kharaty, Catherine M. Phillips

**Affiliations:** 1School of Public Health, Physiotherapy and Sports Science, University College Dublin, Belfield, 4 Dublin, Ireland; 2Institute of Food and Health, School of Agriculture and Food Science, University College Dublin, Belfield, 4 Dublin, Ireland

**Keywords:** plant-based diets, plant-based dietary indices, lipids, lipoproteins, inflammation, biomarkers, cardiovascular disease, atherosclerosis, diet quality, sustainable diets

## Abstract

Plant-based diets (PBDs) are becoming increasingly popular. Thus far, the literature has focused on their association with lipid profiles, with less investigation of lipoprotein and inflammatory profiles. Because pro-atherogenic lipid, lipoprotein, and inflammatory processes may facilitate the development of atherosclerosis, understanding the relation between PBDs and these processes is important to inform risk mitigation strategies. Therefore, the objective of this paper was to review the literature on PBDs and lipid, lipoprotein, and inflammatory biomarkers of cardiovascular disease (CVD). A structured literature search was performed, retrieving 752 records, of which 43 articles were included. Plant-based diets generally associated with favourable lipid and lipoprotein profiles, characterised by decreased total cholesterol, low-density lipoprotein cholesterol, and apolipoprotein B concentrations, and less low-grade inflammation, characterised by decreased C-reactive protein concentrations. Effect sizes from PBD interventions were greatest compared to habitual dietary patterns, and for non-low-fat vegan and tightly controlled dietary interventions. Associations between PBD indices and the reviewed biomarkers were less consistent. Findings are discussed with reference to the literature on PBDs and PBD indices and CVD risk, the associations between specific plant food groups and CVD outcomes and the reviewed biomarker outcomes, and the potential mechanisms underpinning associations between PBDs and reduced CVD risk.

## 1. Introduction

Plant-based diets (PBDs) can be defined as mostly or exclusively plant-based dietary patterns, such as vegan or vegetarian diets. These dietary patterns have shown favourable associations with cardiovascular disease (CVD) risk [1,2], and environmental impacts [3,4,5,6], compared to more animal-based diets. As such, the most recent guidelines from the European Society of Cardiology [7], the American Heart Association (AHA) [8], and the American Society for Preventive Cardiology [9] endorse a more plant-based dietary pattern for more favourable cardiovascular health, while also acknowledging inherent environmental benefits. Improving our understanding of the aetiological factors that may mediate atherogenesis is crucial. Therefore, investigating associations between PBDs and CVD risk, with a view to more effectively mitigating CVD risk, is warranted.

In terms of modifiable CVD risk factors, lipid and lipoprotein profiles are of major importance. Dyslipidaemia is characterised by higher plasma concentrations of total cholesterol (TC), low-density lipoprotein cholesterol (LDL-C), and/or triglyceride (TG), or reduced high-density lipoprotein cholesterol (HDL-C) concentrations [10]. In the INTERHEART study, a large case–control study of approximately 30,000 individuals across 52 countries, dyslipidaemia was the single most important risk factor for first myocardial infarction (MI) [11]. Of the components of dyslipidaemia, LDL-C concentrations are the target of therapy as a consistent line of evidence has demonstrated that when lowered, a dose-dependent, log-linear reduction in coronary heart disease (CHD) risk is observed [12]. This is because LDL-C concentrations approximate total LDL particle concentration, or number of apolipoprotein B (apoB) particles, which can enter the arterial intima and cause atherosclerosis [12,13]. Further, the lipoprotein particle subclass profile may further affect cardiometabolic risk, with a pro-atherogenic phenotype characterised by a preponderance of small, dense LDL [14,15,16], small HDL [16,17], and large very-low-density lipoprotein (VLDL) particle concentrations [16].

Inflammation also plays an important role in the initiation and progression of atherosclerosis [18]. During atherogenesis, an innate immune response is initiated by endothelial cells, forming inflammatory cytokines such as interleukin-6 (IL-6) and tumour necrosis factor alpha (TNF-α) [19]. As low-grade inflammation accompanies all stages of atherosclerosis from the onset to overt disease, elevated levels of circulating inflammatory biomarkers are often used to predict future CVD risk [20]. C-reactive protein (CRP) is the most established of these [21], and is highly predictive of CVD endpoints in long-term prospective cohort studies [22].

To our knowledge, no review has comprehensively characterised the relationship between PBDs and lipid, lipoprotein, and inflammatory biomarkers of CVD, and explored how these biomarkers may mediate atherogenesis. Therefore, this review paper aims to comprehensively summarise the current literature on the relationship between PBDs and these biomarkers.

## 2. Materials and Methods

This is a review of the current literature on the relationships between PBDs and lipid, lipoprotein, and inflammatory biomarkers of CVD.

### 2.1. Search Process

A literature search was performed in September 2022 using three databases: PubMed, Scopus and CINAHL Plus. Key concepts were identified using a Population, Intervention/Exposure, Comparison, Outcomes, and Study Type, (PICOS) framework, which informed the eligibility criteria (Table 1). The literature search strategy is presented in Table 2. Rationale for the eligibility criteria is given by PICOS:**Population:** The chosen population was adults aged 18 years or older because primordial prevention of atherosclerosis and CVD is important and should be prioritised for adults of all ages. In addition, dietary effects on biomarkers of CVD are not age dependent.**Intervention/Exposure:** Only studies looking at PBDs administered as an intervention or those habitually following PBDs (i.e., vegans or vegetarians), or PBD scores/indices as measured by food frequency questionnaire, were eligible for inclusion because the objective of this review is to summarise the literature on plant-based dietary patterns and biomarkers of CVD. Vegan diets were defined as exclusively PBDs, whereas vegetarian diets were defined as consisting of plant foods, while permitting any amount of dairy and/or eggs, and trace amounts of meat and/or fish (<1 serving/d), so as to include cohort studies where the vegetarian groups consumed negligible amounts of either meat or fish.**Comparison:** Intervention and cohort studies that compared PBDs versus other dietary patterns were included to highlight differences between them. PBD scores/indices, compared by quintiles or assessed by continuous measures, were included to observe the effects of eating a more or less PBD, and not necessarily a fully PBD, on established biomarkers of CVD.**Outcomes:** Lipid, lipoprotein, and inflammatory outcomes were eligible for inclusion to ensure that the associations between PBDs, administered as an intervention or followed habitually, and important biomarkers of CVD were captured.**Study type:** Low-quality study types, e.g., case reports/series were excluded.

### 2.2. Data Extraction

After database searching (and duplicate exclusion), 560 records were identified and assessed for eligibility. After full-text screening, a total of 43 articles met the eligibility criteria and relevant data was extracted and tabulated. For cohort and cross-sectional data, results from fully adjusted analyses were included, where possible. A full breakdown of the search and screening process is shown in Figure 1.

## 3. Results

### 3.1. Randomised Controlled Trials of Plant-Based Diets and the Lipid Profile

Table 3 shows results from 27 articles describing randomised controlled trials (RCTs) investigating the association between PBDs and lipid profiles. Trials included individuals living with chronic conditions such as overweight, obesity, and/or type 2 diabetes mellitus (T2DM), were most often conducted in the USA, and investigated a range of different plant-based dietary interventions.

#### 3.1.1. Vegan Dietary Interventions and the Lipid Profile

Intervention trials administering a low-fat vegan diet generally showed decreased TC, LDL-C, and HDL-C concentrations, but increased TG concentrations, when compared to usual-diet control groups [23,24,25,26,27]. Similar findings (including increased very-low-density lipoprotein cholesterol (VLDL-C) concentrations) were observed for 62 overweight individuals when consuming a low-fat vegan diet compared to a Mediterranean diet in a crossover trial [28]. However, when compared to a diet recommended by the American Diabetes Association (ADA), a low-fat vegan diet intervention in individuals with T2DM showed no significant differences in TC, non-HDL-C, LDL-C, HDL-C, VLDL-C, or TG concentrations after an initial 22-week [29], and subsequent 74-week follow-up [30]. There were significant differences in analyses adjusted for medication changes, however, where the low-fat vegan diet group showed significantly decreased TC, non-HDL-C, and LDL-C concentrations compared to the ADA-diet group [29,30]. Two further intervention trials of a low-fat vegan diet in individuals with T2DM failed to observe significant differences in most lipid outcomes when compared to a portion-controlled or low-fat non-vegetarian diet [31,32]. Other interventions employing different iterations of a vegan diet, including extreme, uncooked (raw) versions, generally showed significant or non-significant decreases in TC, LDL-C, and HDL-C concentrations compared to non-vegetarian control groups, whereas the direction of changes in TG concentrations were less consistent across trials [33,34,35,36].

The greatest changes in lipid concentrations were observed in tightly controlled trials. In a metabolically controlled crossover trial including 20 young adults, an ad libitum low-fat plant-based (vegan) diet showed significantly decreased TC, LDL-C, and HDL-C concentrations, but increased TG concentrations, compared to an animal-based ketogenic diet intervention [37]. In another, a portfolio (vegan) dietary pattern showed non-inferiority in lipid-lowering potential to 20 mg/d of lovastatin therapy in 34 hyperlipidaemic adults, and large, significant reductions in TC and LDL-C concentrations compared to a low-saturated-fat diet (control) group [38]. Finally, a metabolically controlled parallel trial (*n* = 44) comparing a low-carbohydrate vegan diet to a low-fat lacto-ovo-vegetarian diet reported decreased TC, LDL-C, HDL-C, and TG concentrations from baseline for both, and significantly decreased TC, LDL-C, and TG concentrations for the low-carbohydrate vegan diet compared to the comparator [39]. These findings persisted at 6-month follow-up, but effect sizes attenuated somewhat under free-living conditions [40].

#### 3.1.2. Vegetarian Dietary Interventions and the Lipid Profile

In studies prescribing either a calorie- and fat-restricted lacto-ovo-vegetarian or non-vegetarian diet in individuals with overweight or obesity, non-significant differences were observed between groups in all lipid outcomes at 6- and 18 months of follow-up [41,42]. However, a low-calorie lacto-ovo-vegetarian diet intervention showed a significant decrease in LDL-C concentrations and increase in TG concentrations compared to a low-calorie Mediterranean diet intervention in a crossover trial including individuals with overweight or obesity [43]. In other crossover trials, vegetarian diet interventions showed decreased TC, LDL-C, and HDL-C concentrations compared to standard USA-style non-vegetarian diets [44,45], or an isocaloric non-vegetarian diet adhering to Nordic recommendations [46]. In a 6-week trial including 173 overweight and pre-menopausal females, a soy-protein and non-soy plant-protein diet intervention showed significantly decreased TC, LDL-C, and HDL-C concentrations compared to an animal protein and/or dairy protein diet intervention [47]. However, in a small study of chronic renal failure patients (*n* = 9), a soy-based low-protein vegetarian diet intervention showed non-significant decreases in TC, LDL-C, and HDL-C concentrations, and increased TG concentrations, compared to an animal-based low-protein diet intervention [48]. In a 4-week intervention including healthy, overweight individuals (*n* = 120), a lacto-ovo-vegetarian diet showed significantly decreased TC and LDL-C concentrations compared to a eucaloric low-fat non-vegetarian diet matched for saturated fat and dietary cholesterol [49], while both diet interventions lowered concentrations from baseline.

#### 3.1.3. Summary of Randomised Controlled Trials Investigating Vegan and Vegetarian Dietary Interventions and the Lipid Profile

In summary, vegan and vegetarian diets tend to decrease TC, LDL-C, and HDL-C concentrations compared to non-vegetarian diets, with effects most evident compared to Western-style diet controls. Increases in TG concentrations were consistently reported for low-fat vegan dietary interventions compared to non-vegetarian diet comparators, however this effect was not observed in non-low-fat vegan dietary interventions, where TG concentrations were often reduced from baseline, and compared to control groups. The largest effect sizes were reported in trials employing non-low-fat vegan dietary interventions, and in the most tightly (metabolically) controlled trials.

**Table 3 nutrients-14-05371-t003:** Randomised Controlled Trials of Plant-Based Diets and Lipid Profiles.

Reference	Country	Population (*n*)	Sex	Age (Years)	Intervention (*n*)	Study Length/Design	Outcomes	* Results	Significance
Acharya et al. [42]	USA	Overweight/obese (143)	M/F	LOV-D: 45.2; STD-D: 43.5	LOV-D (64) vs. STD-D (79)	6 months (parallel)	TC, LDL-C, HDL-C, TGs,	Changes from baseline (%): LOV-D: TC: −4.7, LDL-C: −6.1, HDL-C: −5.5, TGs: −3.8. STD-D: TC: −1.2, LDL-C: −4.2, HDL-C: −3.0, TGs: −1.26	Both diets lowered lipid outcomes from baseline, but differences between diets were non-significant (*p* > 0.05)
Ågren et al. [34]	Finland	Rheumatoid arthritis (29)	M/F	VG: 49.0; NVD: 53.0	VG (16) vs. NVD (13)	3 months (parallel)	TC, LDL-C, HDL-C, TGs	TC: −0.94; LDL-C: −0.74; HDL-C: −0.16; TGs: −0.11	*p* < 0.001 for TC and LDL-C; *p* > 0.05 (ns) for HDL-C and TGs
Barnard et al. [26]	USA	Healthy pre-menopausal women (35)	F	All: 36.1	LFVG vs. usual diet + placebo pill	5 menstrual cycles for each arm (crossover)	TC, LDL-C, HDL-C, VLDL-C, TGs	TC: −0.54; LDL-C: −0.3; HDL-C: −0.2; VLDL-C: +0.08; TGs: +0.18	*p* < 0.001 for all but TGs (*p* < 0.01)
Barnard et al. [29]	USA	T2DM (99)	M/F	LFVG: 56.7; ADA: 54.6	LFVG (49) vs. ADA-recommended diet (50)	22 weeks (parallel)	TC, non-HDL-C, LDL-C, HDL-C, VLDL-C, TGs	ITT analysis: TC: −0.09; non-HDL-C: −0.05; LDL-C: −0.03; HDL-C: −0.05; VLDL-C: +0.03; TGs: −0.04; Medication-change-adjusted analysis: TC: −0.38; non-HDL-C: −0.29; LDL-C: −0.31; HDL-C: −0.08; VLDL-C: +0.01; TGs: +0.01	ns (*p* > 0.05) difference between groups for all outcomes in ITT analysis; significantly lower TC (*p* = 0.01), non-HDL-C (*p* = 0.05) and LDL-C (*p* = 0.02) in analyses adjusted for medication changes.
Barnard et al. [30]	USA	T2DM (99)	M/F	LFVG: 56.7; ADA: 54.6	LFVG (49) vs. ADA-recommended diet (50)	74 weeks (parallel)	TC, non-HDL-C, LDL-C, HDL-C, VLDL-C, TGs	ITT analysis: TC: −0.18; non-HDL-C: −0.21; LDL-C: −0.11; HDL-C: +0.01; VLDL-C: −0.02; TGs: −0.29; Medication-change-adjusted analysis: TC: −0.35; non-HDL-C: −0.35; LDL-C: −0.26; HDL-C: −0.01; VLDL-C: −0.05; TGs: −0.32	ns (*p* > 0.05) difference between groups for all outcomes in ITT analysis; significantly lower TC (*p* = 0.01), non-HDL-C (*p* = 0.02) and LDL-C (*p* = 0.03) in analyses adjusted for medication changes.
Barnard et al. [31]	USA	T2DM (45)	M/F	LFVG: 61.0; portion-controlled: 61.0	LFVG (21) vs. portion-controlled group (24)	20 weeks (parallel)	TC, LDL-C, HDL-C, TGs	TC: + 0.21; LDL-C: +0.02; HDL-C: +0.03; TGs: +0.52	ns (*p* > 0.05) difference between groups for all outcomes
Barnard et al. [28]	USA	Overweight (62)	M/F	LFVG: 58.3; MD: 56.6	LFVG (30) vs. MD (32)	36 weeks: 16 weeks × 2 (crossover) with a 4-week washout in between	TC, LDL-C, HDL-C, TGs, VLDL-C	TC: −0.29; LDL-C: −0.28; HDL-C: −0.11; TGs: +0.23; VLDL-C: +0.11	Treatment effect: *p* = 0.04 for TC and LDL-C; *p* = 0.009 for HDL-C; *p* = 0.01 for TGs and VLDL-C
Burke et al. [41]	USA	Overweight/obese (176)	M/F	LOV-D: 45.4; STD-D: 43.3	LOV-D (90) vs. STD-D (96)	18 months: 12-month intervention, 6-month maintenance phase (parallel)	TC, TGs	Changes given in %: STD-D baseline to 18 months (preference group yes/no): TC: −1.4/+2.5; TGs: +1.0/−6.7; LOV-D (preference group yes/no): TC: +1.0/−0.1; TGs: +8.6/−5.5	ns (*p* > 0.05) difference between groups for all outcomes
Cooper et al. [44]	USA	Healthy (15)	M/F	All: 28.0	LOV vs. typical USA diet	6 weeks: 3 weeks × 2 (crossover)	TC, LDL-C, HDL-C, TGs	TC: −0.52; LDL-C: −0.41; HDL-C: −0.10; TGs: −0.02	*p* < 0.05 for TC; *p* < 0.025 for LDL-C; ns (*p* > 0.05) for other outcomes
Djekic et al. [46]	Sweden	Overweight (31)	M/F	LOV: 67.0; NVD: 68.0	Isocaloric LOV (16) vs. NVD (15) [both adhering to Nordic Recommendations]	12 weeks: 4 weeks × 2 (crossover) with a 4-week washout in between	TC, LDL-C, HDL-C, TGs	TC: −0.13; LDL-C: −0.10; HDL-C: −0.03; TGs: +0.06	*p* = 0.01 for TC, *p* = 0.02 for LDL-C; ns (*p* > 0.05) for all other outcomes
Elkan et al. [35]	Sweden	Rheumatoid arthritis (66)	M/F	VG: 50.0; NVD: 50.8	VG gluten-free (38) vs. NVD (28)	12 months (parallel)	TC, LDL-C, HDL-C, TGs	TC: −1.2; LDL-C: −1.1; HDL-C: 0.0; TGs: 0.0	*p* < 0.001 for LDL-C; no significance test reported for difference between diet groups for all other outcomes
Ferdowsian et al. [27]	USA	Overweight and T2DM (113)	M/F	21 to 65	LFVG (68) vs. usual-diet control (45)	22 weeks (parallel)	TC, LDL-C, HDL-C, TGs	TC: −0.21; LDL-C: −0.08; HDL-C: −0.10; TGs: −0.20	*p* = 0.002 for HDL-C; ns (*p* > 0.05) for all other outcomes
Gardner et al. [49]	USA	Healthy and overweight (120)	M/F	LFLOV: M: 48.0 & F: 48.0; LFD: M: 49.0 & F: 46.0	LFLOV (59) vs. Eucaloric LFD (61)	4 weeks (parallel)	TC, LDL-C, HDL-C, TGs	TC: −0.22, LDL-C: −0.18, HDL-C: −0.04; TGs: −0.01	Lower TC (*p* = 0.01) and LDL-C (*p* = 0.02); ns differences for HDL-C and TGs
Gonciulea and Sellmeyer [47]	USA	Overweight and pre-menopausal (173)	F	APD: 62.7; DPD: 64.5; NSPD: 62.2; SPD: 64.6	Energy- and protein-matched APD vs. DPD vs. NSPD vs. SPD	6 weeks (parallel)	TC, LDL-C, HDL-C, TGs	SPD vs. APD: TC: −0.56; LDL-C: −0.43; HDL-C: −0.14; TGs: +0.06; SPD vs. DPD: TC: −0.77; LDL-C: −0.69; HDL-C: −0.16; TGs: +0.14; NSPD vs. APD: TC: −0.35; LDL-C: −0.26; HDL-C: −0.09; TGs: +0.05; NSPD vs. DPD: TC: −0.56; LDL-C: −0.49; HDL-C: −0.11; TGs: +0.13	SPD vs. APD: *p* < 0.001 for TC and LDL-C, *p* = 0.008 for HDL-C; SPD vs. DPD: *p* < 0.001 for TC and LDL-C, *p* = 0.003 for HDL-C; NSPD vs. APD: *p* = 0.02 for TC, *p* = 0.04 for HDL-C; NSPD vs. DPD: *p* = 0.003 for TC, *p* = 0.005 for LDL-C, *p* = 0.05 for HDL-C; all other results ns (*p* > 0.05)
Hall et al. [37]	USA	Overweight (20)	M/F	All: 29.9	LFPBD vs. ABKD	4 weeks: 2 weeks × 2 (crossover)	TC, LDL-C, HDL-C, TGs	TC: −1.11; LDL-C: −0.72; HDL-C: −0.25; TGs: +0.34	*p* < 0.001 for all
Hunt et al. [45]	USA	Healthy (21)	F	All: 33.2	LOV vs. NVD	8 weeks: 4 weeks × 2 (crossover)	TC, LDL-C, HDL-C, TGs	TC: −0.37; LDL-C: −0.25; HDL-C: −0.14; TGs: +0.06	*p* = 0.001 for TC and LDL-C; *p* = 0.05 for HDL-C; *p* > 0.05 (ns) for TGs
Jenkins et al. [38]	Canada	Hyperlipidaemic (34)	M/F	All: 58.4	Statin vs. Portfolio Diet vs. low saturated fat control diet	3 × 1 month (crossover) intervention periods with a 2-to-6-week washout period between	TC, LDL-C, HDL-C, TGs	TC: −1.12; LDL-C: −0.99; HDL-C: +0.04; TGs: −0.38	*p* < 0.005 for TC and LDL-C; ns (*p* > 0.05) for HDL-C and TGs; results were non-significantly different for all included outcomes
Jenkins et al. [39]	Canada	Overweight and hyperlipidaemia (44)	M/F	LCPBD: 56.1; LFLOV: 57.8	LCPBD (22) vs. LFLOV (22)	1-month parallel, metabolically controlled study	TC, LDL-C, HDL-C, TGs	LCPBD: TC: −1.34; LDL-C: −0.96; HDL-C: −0.05; TGs: −0.86; LFLOV: TC: −0.83; LDL-C: −0.57; HDL-C: −0.08; TGs: −0.45	LCPBD had significantly lower TC (*p* = 0.001), LDL-C (*p* = 0.002), and TGs (*p* = 0.02) vs. LFLOV; ns (*p* > 0.05) changes in HDL-C between groups
Jenkins et al. [40]	Canada	Overweight and hyperlipidaemia (39)	M/F	LCPBD: 57.6; LFLOV: 55.3	LCPBD (20) vs. LFLOV (19)	6 months (parallel)	TC, LDL-C, HDL-C, TGs	LCPBD: TC: −0.66; LDL-C: −0.47; HDL-C: +0.04; TGs: −0.73; LFLOV: TC: −0.26; LDL-C: 0.00; HDL-C: −0.01; TGs: −0.45	LCPBD had significantly lower TC (*p* < 0.001), LDL-C (*p* < 0.001), and TGs (*p* = 0.005) vs. LFLOV; ns (*p* > 0.05) changes in HDL-C between groups
Kahleova et al. [23]	USA	Overweight (222)	M/F	LFVG: 53.0; Control: 57.0	LFVG (117) vs. usual diet control (105)	16 weeks (parallel)	TC, LDL-C, HDL-C, TGs	TC: −0.6; LDL-C: −0.5; HDL-C: −0.01; TGs +0.20	*p* < 0.001 for TC and LDL-C; *p* = 0.02 for TGs; ns difference for HDL-C
Ling et al. [36]	Finland	Healthy (18)	M/F	VG: 48.0; NVD: 37.5	Uncooked VG (including fermented foods) vs. mixed NVD	4 weeks (parallel)	TC, LDL-C, HDL-C, TGs	TC: −0.77; LDL-C: −0.74; HDL-C: −0.09; TGs: −0.31	No significance tests were conducted between groups. The VG diet significantly lowered TC (*p* < 0.001), LDL-C (*p* < 0.001), HDL-C (*p* < 0.01), and TGs (*p* < 0.05) vs. baseline values.
Mishra et al. [24]	USA	Overweight and T2DM (291)	M/F	LFVG: 44.3; Control: 46.1	LFVG (142) vs. usual-diet control (149)	18 weeks (parallel)	TC, LDL-C, HDL-C, TGs	TC: −0.21; LDL-C: −0.19; HDL-C: −0.07; TGs: +0.13	*p* < 0.01 for TC, LDL-C, and HDL-C; *p* < 0.05 for TGs
Nicholson et al. [32]	USA	T2DM (11)	M/F	LFVG: 51; Conventional LFD: 60	LFVG (7) vs. conventional LFD (4)	12 weeks (parallel)	TC, HDL-C, TGs	TC: 0.00; HDL-C: −0.18; TGs: +0.19	*p* < 0.05 for HDL-C, ns (*p* > 0.05) for TC and TGs
Shah et al. [33]	USA	Coronary artery disease (100)	M/F	VG: 63.0; AHA: 59.5	VG (50) vs. AHA-recommended diet (50)	8 weeks (parallel)	TC, non-HDL-C; LDL-C, HDL-C, TGs	TC: −0.13; non-HDL-C: 0.00; LDL-C: −0.21; TGs: +0.11	ns (*p* > 0.0015) differences between groups for all outcomes using linear regression analysis (Bonferroni correction applied)
Sofi et al. [43]	Italy	Overweight/obese with elevated TC or LDL-C or TGs or glucose (118)	M/F	LCLOV: 49.5; LCMD: 52.0	Isocaloric hypocaloric LCLOV vs. LCMD	6 months: 3 months × 2 (crossover)	TC, LDL-C, HDL-C, TGs	TC: −0.14; LDL-C: −0.24 mmol/L; HDL-C: −0.03; TGs: +0.14	*p* ≤ 0.01 for LDL-C and TGs; ns (*p* > 0.05) for other outcomes
Soroka et al. [48]	Israel	Chronic renal failure (9)	M/F	30 to 85	Soya-based vegetarian low-protein diet vs. animal-based low-protein diet	12 months: 6 months × 2 (crossover)	TC, LDL-C, HDL-C, TGs	TC: −0.03; LDL-C: −0.10; HDL-C: −0.07; TGs: +0.56	ns (*p* > 0.05) for all comparisons
Wright et al. [25]	New Zealand	Overweight/obese with comorbidities (49)	M/F	All: 56.0	LFVG (25) vs. control (normal GP care; 24)	6 months (parallel)	TC, LDL-C, HDL-C, TGs	TC: −0.5; LDL-C: −0.4; HDL-C: −0.2; TGs: +0.2; Excluding dropouts: LFVG vs. control for TC: −0.56	*p* = 0.001 for HDL-C; ns (*p* > 0.05) for all other differences in outcomes; *p* = 0.05 for differences in TC excluding dropouts

**Abbreviations:** ABKD: animal-based ketogenic diet; ADA: American Diabetes Association; AHA: American Heart Association; APD: animal protein diet; DPD: dairy protein diet; F: female; GP: general practitioner; HDL-C: high-density lipoprotein cholesterol; ITT: intention to treat; LCLOV: low-calorie lacto-ovo-vegetarian diet; LCMD: low-calorie Mediterranean diet; LCPBD: low-carbohydrate plant-based diet; LDL-C: low-density lipoprotein cholesterol; LFD: low-fat diet; LFLOV: low-fat lacto-ovo-vegetarian diet; LFPBD: low-fat plant-based diet; LFVG: low-fat vegan diet; LOV-D: calorie- and fat-restricted lacto-ovo-vegetarian diet; LOV: lacto-ovo-vegetarian diet; M: male; MD: Mediterranean diet; NVD: non-vegetarian diet; ns: non-significant; NSPD: non-soy plant protein diet; PBD: plant-based diet; SPD: soy protein diet; STD-D: standard calorie- and fat-restricted diet; T2DM: type 2 diabetes mellitus; TC: total cholesterol; TGs: triglycerides; VG: vegan diet; VLDL-C: very-low-density lipoprotein cholesterol. * Results are presented as the difference between interventions (PBD vs. comparison) for all trials except for Jenkins et al. [39,40]. All lipid measurements are given as mmol/L unless otherwise stated. Age is reported as mean, median, or range.

### 3.2. Cohort Studies of Plant-Based Diets and the Lipid Profile

Table 4 shows results from 2 prospective cohort studies investigating the association between those consuming PBDs and lipid profiles. Both cohorts included healthy middle-aged individuals from Taiwan, reporting a greater likelihood for vegans to have low HDL-C concentrations compared to non-vegetarians and pesco-vegetarians in one study [50], and non-significant differences for vegans, lacto-ovo-vegetarians, and lacto-vegetarians compared to non-vegetarians for likelihood of high TC, high LDL-C, low HDL-C, or high TG concentrations in another [51].

### 3.3. Randomised Controlled Trials of Plant-Based Diets and the Lipoprotein Profile

Table 5 shows results from 9 RCTs investigating the association between PBDs and lipoprotein profiles. Studies featured a majority of North American cohorts, and included healthy individuals, individuals with overweight and/or hyperlipidaemia, or individuals with CHD.

#### 3.3.1. Vegan Dietary Interventions and the Lipoprotein Profile

Five trials administered a vegan dietary intervention and measured apoB concentrations, reporting large, significant reductions compared to baseline and comparison diet groups [36,37,38,39,40]. Hall et al. [37], Jenkins et al. [38], and Ling et al. [36] all reported similarly large effect sizes compared to non-vegetarian diets, despite each trial including different iterations of non-vegetarian diets, that is, an animal-based ketogenic diet, a low-saturated-fat non-vegetarian diet, and a mixed non-vegetarian diet, respectively. In another trial from Jenkins et al. [39], a low-carbohydrate vegan dietary intervention conducted under metabolically controlled conditions showed a substantial decrease in apoB concentrations compared to baseline, and significantly decreased concentrations compared to a low-fat lacto-ovo-vegetarian diet comparator. These results remained significant after a 6-month follow-up, however the effect size attenuated somewhat [40].

Only 2 trials investigated the lipoprotein subclass profile. Hall et al. [37] reported significantly decreased total LDL and HDL, small and medium LDL, and small and large HDL particle concentrations, but significantly increased total VLDL, large LDL, and medium HDL particle concentrations, compared to an animal-based ketogenic diet intervention. Shah et al. [33] compared a vegan dietary intervention to an AHA-recommended dietary intervention, reporting non-significant decreases in total LDL and large HDL particle concentrations, LDL particle size, and HDL particle size, and non-significant increases in total HDL and small LDL particle concentrations, and VLDL particle size

#### 3.3.2. Vegetarian Dietary Interventions and Apolipoprotein B Concentrations

Five trials administered a vegetarian dietary intervention and measured apoB concentrations, reporting significant decreases from baseline in all, and either borderline or non-significant decreases compared to non-vegetarian diet comparators [39,40,44,45,46]. Effect sizes were relatively small in comparison to vegan dietary interventions.

#### 3.3.3. Summary of Vegan and Vegetarian Dietary Interventions and the Lipoprotein Profile

In all trials that measured apoB, PBD interventions typically showed significantly decreased concentrations compared to non-vegetarian comparator groups. Effect sizes were largest in tightly controlled trials and in vegan dietary interventions. There is a dearth of trials investigating PBD interventions and the lipoprotein subclass profile, highlighting the need for further investigation.

### 3.4. Randomised Controlled Trials of Plant-Based Diets and the Inflammatory Profile

Table 6 shows results from 12 RCTs investigating the association between PBDs and inflammatory profiles. Trials included individuals living with chronic conditions such as overweight, obesity, or T2DM, were most often conducted in the USA, and investigated a range of different plant-based dietary interventions.

#### 3.4.1. Vegan Dietary Interventions and the Inflammatory Profile

In a crossover trial including overweight individuals (*n* = 20), a low-fat plant-based (vegan) dietary intervention found a significant decrease in high-sensitivity C-reactive protein (hsCRP) concentrations compared to an animal-based ketogenic diet intervention [37]. However, when compared to an ADA-recommended diet, a low-fat vegan diet intervention showed a non-significant decrease in CRP concentrations in individuals with T2DM [30]. Another intervention examining a vegan diet showed a significant decrease in hsCRP concentrations and a non-significant increase in white blood cell (WBC) count and neutrophil-to-lymphocyte ratio compared to an AHA-recommended diet [33]. However, when compared to a meat-rich diet, a vegan diet intervention showed significant decreases in leukocyte and monocyte concentrations and non-significant decreases in hsCRP and lymphocyte concentrations [52]. A further trial showed a decrease in CRP concentrations in individuals adhering to a vegan diet compared to a non-vegan diet control group, in individuals with rheumatoid arthritis [35]. Two further trials administering a low-carbohydrate vegan diet in a Canadian cohort of overweight and hyperlipidaemic participants showed either significant or non-significant decreases in hsCRP compared to a low-fat lacto-ovo-vegetarian diet [39,40].

#### 3.4.2. Vegetarian Dietary Interventions and the Inflammatory Profile

In a study prescribing a calorie- and fat-restricted lacto-ovo-vegetarian diet in overweight or obese individuals, a non-significant increase in adiponectin was observed for the lacto-ovo-vegetarian group [42]. In a crossover trial, a lacto-ovo-vegetarian diet intervention showed a non-significant decrease in hsCRP, compared to an isocaloric non-vegetarian diet adhering to Nordic recommendations [46]. A decrease in WBC count was shown in a small (*n* = 12) 12-week study including USA males for the lacto-ovo-vegetarian diet intervention compared to the beef-containing diet intervention [53]. Another crossover intervention examining a low-calorie lacto-ovo-vegetarian diet showed a non-significant increase in IL-6 and TNF-α concentrations and WBC count compared to a low-calorie Mediterranean diet intervention in individuals with overweight or obesity [43]. A further trial reported decreased leptin and resistin concentrations, a decreased leptin-to-adiponectin ratio, and increased adiponectin concentrations, for a lacto-ovo-vegetarian diet intervention compared to a Mediterranean diet intervention; however, all results were non-significant [54]. In trials from Jenkins et al. [39,40], while the low-fat lacto-ovo-vegetarian diet intervention decreased CRP concentrations to a lesser extent than the low-carbohydrate vegan diet intervention, concentrations were significantly reduced from baseline.

#### 3.4.3. Summary of Vegan and Vegetarian Dietary Interventions and the Inflammatory Profile

The majority of vegan intervention studies showed either significant or non-significant decreases in inflammatory biomarkers, the most common being CRP/hsCRP concentrations, from baseline and compared to non-vegetarian diet interventions. Vegetarian diet interventions showed less clear evidence of reduced inflammation compared to non-vegetarian diet comparators, with results mixed. Decreases in inflammatory biomarkers were less clear when PBDs were compared to other established healthy diets, e.g., Mediterranean diet.

**Table 6 nutrients-14-05371-t006:** Randomised Controlled Trials of Plant-Based Diets and Inflammatory Biomarkers.

Reference	Country	Population (*n*)	Sex	Age (Years)	Intervention (*n*)	Study Length (Design)	Outcomes	* Results	Significance
Acharya et al. [42]	USA	Overweight/obese (143)	M/F	LOV-D: 45.2; STD-D: 43.5	LOV-D (64) vs. STD-D (79)	6 months (parallel)	Adiponectin (µg/mL)	Changes from baseline (%): LOV-D: +9.4; STD-D: +7.2 (difference: +2.2)	ns (*p* = 0.45) difference between groups
Barnard et al. [30]	USA	T2DM (99)	M/F	LFVG: 56.7; ADA: 54.6	LFVG (49) vs. ADA-recommended diet (50)	74 weeks (parallel)	CRP (mg/L)	ITT analysis: −5.0	ns (*p* = 0.65) difference between groups
Dinu et al. [54]	Italy	Healthy (118)	M/F	LOV: 50.5; MD: 52	LOV (54) vs. MD (53)	3 months	Leptin (ng/mL), adiponectin (µg/mL), LAR, resistin (ng/mL)	LOV: leptin: −0.58, adiponectin: +0.49, LAR: −0.12, resistin: −0.12; MD: leptin: −1.35 (difference: +0.77), adiponectin: +0.45 (difference: +0.04), LAR: −0.17 (difference +0.05), resistin: +0.04 (difference: −0.16)	ns (*p* > 0.05) difference between groups for all outcomes
Djekic et al. [46]	Sweden	Overweight (31)	M/F	LOV: 67.0; NVD: 68.0	Isocaloric LOV (16) vs. NVD (15) [both adhering to Nordic Recommendations]	12 weeks: 4 weeks × 2 (crossover) with a 4-week washout in between	hsCRP (mg/L)	Difference: −0.09	ns (*p* = 0.6) difference between groups
Elkan et al. [35]	Sweden	Rheumatoid arthritis (66)	M/F	VG: 50.0; NVD: 50.8	VG gluten-free (38) vs. NVD (28)	12 months (parallel)	CRP (mg/L)	VG: −8; NVD: −10 (difference: +2)	no significance test reported for difference between diet groups
Hall et al. [37]	USA	Overweight (20)	M/F	All: 29.9	LFPBD vs. ABKD	4 weeks: 2 weeks × 2 (crossover)	hsCRP (mg/L)	LFPBD: −0.9; ABKD: 0 (difference: −0.9)	*p* = 0.003
Jenkins et al. [39]	Canada	Overweight with hyperlipidaemia (44)	M/F	LCPBD: 56.1; LFLOV: 57.8	LCPBD (22) vs. LFLOV (22)	1-month parallel, metabolically controlled study	hsCRP (mg/L)	LCPBD: −0.89; LFLOV: −0.69 (difference: −0.2)	ns (*p* = 0.66) difference between groups
Jenkins et al. [40]	Canada	Overweight with hyperlipidaemia (39)	M/F	LCPBD: 57.6; LFLOV: 55.3	LCPBD (20) vs. LFLOV (19)	6 months (parallel)	hsCRP (mg/dL)	LCPBD: −0.4; LFLOV: −0.2 (difference: −0.2)	ns (*p* = 0.082) difference between groups
Lederer et al. [52]	Germany	Healthy (53)	M/F	VG: 33.2; OD: 29.9	VG (26) vs. meat-rich diet (27)	4 weeks (w/ pre-treatment controlled mixed diet for 1 week)	Leukocytes (thousands/μL), monocytes (thousands/μL), hsCRP (mg/dL), lymphocytes (thousands/μL)	VG: hsCRP: −0.2, leukocytes: −0.6, lymphocytes: −35.7, monocytes: −0.03; Meat-rich diet: CRP: +0.2 (difference: −0.04), leukocytes: 0 (difference: −0.06), lymphocytes: +0.8 (difference: −35.78), monocytes: +0.03 (difference: −0.06)	Leukocytes (*p* = 0.003), monocytes (*p* = 0.032); ns (*p* > 0.05) differences for all other outcomes
Shah et al. [33]	USA	Coronary artery disease (100)	M/F	VG: 63.0; AHA: 59.5	VG (50) vs. AHA-recommended diet (50)	8 weeks (parallel)	hsCRP (mg/L), WBC count (K/μL), NLR	Adjusted *β* for VG vs. AHA-recommended diet (as reference): hsCRP: 0.67, WBC count: 1.06, NLR: 1.20	hsCRP (*p* = 0.02); ns (*p* > 0.05) differences for all other outcomes
Sofi et al. [43]	Italy	Overweight or obesity with elevated TC or LDL-C or TGs or glucose (118)	M/F	LCLOV: 49.5; LCMD: 52.0	Isocaloric hypocaloric LCLOV vs. LCMD	6 months: 3 months × 2 (crossover)	WBC count (× 10³/mm³), IL-6 (pg/mL), TNF-α (pg/mL)	LOV: WBC count: +0.16, IL-6: +0.07, TNF-α: +0.45; MD: WBC count: −0.09 (difference: +0.25), IL-6: −0.09 (difference: +0.16), TNF-α: −0.34 (difference: +0.79)	ns (*p* > 0.05) differences for all outcomes
Wells et al. [53]	USA	Healthy (21)	M	59 to 78	LOV (10) vs beef-containing diet (11)	12 weeks (w/ pre-treatment vegetarian diet for 2 weeks)	WBC count (10⁹/L)	LOV: −0.2; Beef-containing diet: +0.5 (difference: −0.7)	no significance test reported for difference between diet groups

**Abbreviations:** ABKD: animal-based ketogenic diet; ADA: American Diabetes Association; AHA: American Heart Association; CRP: C-reactive protein; F: female; hsCRP: high-sensitivity C-reactive protein; IL-6: interleukin 6; ITT: intention to treat; LAR: leptin-to-adiponectin ratio; LCLOV: low-calorie lacto-ovo-vegetarian diet; LCMD: low-calorie Mediterranean diet; LCPBD: low-carbohydrate plant-based diet; LDL-C: low-density lipoprotein cholesterol; LFLOV: low-fat lacto-ovo-vegetarian diet; LFPBD: low-fat plant-based diet; LFVG: low-fat vegan diet; LOV-D: calorie- and fat-restricted lacto-ovo-vegetarian diet; LOV: lacto-ovo-vegetarian diet; M: male; MD: Mediterranean diet; NLR: neutrophil-to-lymphocyte ratio; NVD: non-vegetarian diet; ns: non-significant; PBD: plant-based diet; STD-D: standard calorie- and fat-restricted diet; T2DM: type 2 diabetes mellitus; TC: total cholesterol; TGs: triglycerides; TNF-α: tumour necrosis factor alpha; VG: vegan diet; WBCs: white blood cells. * Results are presented for each intervention (i.e., PBD; comparison), and then the difference between interventions (PBD vs. comparison), where possible. Age is reported as mean, median, or range.

### 3.5. Plant-Based Diet Indices and Lipid and Inflammatory Profiles

A number of PBD scores, or indices, have been created to assess how adherence to PBDs may associate with intermediate outcomes of disease, such as lipid, lipoprotein, or inflammatory biomarkers, or hard health outcomes of disease, such as CVD. As such, they do not assess categorical dietary patterns, rather, they look at adherence to plant-based (vegan) dietary patterns. The most established are the PBD indices (PDIs), which include an overall PDI, a healthful PDI (hPDI), and an unhealthful PDI (uPDI). The PDI weighs all plant foods positively and all animal foods negatively, whereas the hPDI and uPDI differentiates by PBD quality: the hPDI weighs only healthy plant foods (whole grains, fruits, vegetables, nuts, legumes, vegetable oils, tea/coffee) positively, and all else negatively; the uPDI weighs only unhealthy plant foods (fruit juices, refined grains, potatoes, sugar-sweetened beverages, sweets/desserts) positively, and all else negatively [55]. For a detailed explanation of the PDIs, see Satija et al. [56].

#### 3.5.1. Prospective Cohort Studies of Plant-Based Diet Indices and the Lipid Profile

Table 7 shows results from 3 prospective cohort studies investigating the association between PBD indices and lipid profiles. Studies featured a majority of Korean cohorts, and included healthy individuals or individuals with overweight or obesity and pre-diabetes.

In healthy middle-aged Korean adults followed up over 8–14 years, greater adherence to an uPDI associated with a significantly increased risk for low HDL-C concentrations, hypertriglyceridaemia, and dyslipidaemia, whereas higher adherence to the PDI and hPDI showed a significantly decreased risk for dyslipidaemia [57,58]. In a secondary analysis of 3-year follow-up data from a European, multi-country RCT [59], adherence to a novel PBD score showed a borderline significant association with decreased yearly changes in LDL-C concentrations.

#### 3.5.2. Cross-Sectional Studies of Plant-Based Diet Indices and the Lipid Profile

Table 8 shows results from 5 cross-sectional studies investigating the association between PBD indices and lipid profiles. Studies featured cohorts from Europe, Iran, and the USA, and included healthy individuals or individuals living with chronic conditions such as overweight or obesity with metabolic syndrome, or chronic kidney disease.

In cross-sectional investigations employing the PDIs, greater adherence to the hPDI and uPDI was associated with significantly increased HDL-C concentrations in a healthy Iranian (*n* = 179) cohort [55], whereas greater adherence to the PDI was associated with significantly increased TC concentrations in a healthy USA cohort (*n* = 3635) [60]. In a Swedish cohort of chronic kidney disease patients (*n* = 418), no significant associations were observed between adherence to the PDI and hyperlipidaemia [61]. In studies employing the pro-vegetarian diet indices, which are similar to the PDIs, greater adherence to the general pro-vegetarian diet index predicted non-significant differences in LDL-C, HDL-C, and TG concentrations in one [62], and significantly increased HDL-C concentrations in another [63]. Greater adherence to the unhealthful pro-vegetarian diet index, either comparing extreme quintiles of adherence or per 5-unit increment in score, predicted significantly decreased HDL-C concentrations and increased TG concentrations in overweight or obese individuals with metabolic syndrome [63].

#### 3.5.3. Summary of Studies Investigating Plant-Based Diet Indices and the Lipid Profile

In limited data from prospective cohort and cross-sectional studies, greater adherence to the PDI and hPDI showed evidence of a more favourable lipid profile, whereas greater adherence to the uPDI showed the opposite. Results are fairly consistent with the interventional and observational data investigating the relationship between PBDs and the lipid profile, presented previously.

**Table 8 nutrients-14-05371-t008:** Cross-Sectional Studies of Plant-Based Diet Indices and Lipid Profiles.

Reference	Country	Population (*n*)	Sex	Age (Years)	Intervention (*n*)	Outcome(s)	Results	Significance
Alvarez-Alvarez et al. [62]	Spain	Overweight/obese with metabolic syndrome (6,874)	M/F	64 to 65	PVG	LDL-C, HDL-C, TGs	Regression *β* coefficient for pro-vegetarian diet index (mmol/L): LDL-C: −0.724 (−1.622, 0.173); HDL-C: −0.039 (−0.328, 0.249); TGs: 1.120 (−0.860, 3.101)	*p* > 0.05 for all
Amini et al. [55]	Iran	Healthy (178)	M/F	67.0	Adherence to PDI, hPDI, uPDI	TC, LDL-C, HDL-C, TGs	T3 vs. T1 for hPDI (HDL-C): +0.11 mmol/L; for uPDI (HDL-C): +0.09 mmol/L	*p* = 0.02 for both; ns differences in all other outcomes
González-Ortiz et al. [61]	Sweden	Chronic kidney disease patients (418)	M	71.0	Adherence to PDI	Hyperlipidaemia (TC > 5.2, TGs > 1.71 or treatment with lipid-lowering medications)	No significant difference across quintiles of PDI adherence in rates of hyperlipidaemia	*p*-trend = 0.82
Oncina-Cánovas et al. [63]	Spain	Overweight/obese with metabolic syndrome (6,439)	M/F	64.5 to 65.7	Adherence to gPVG, hPVG, uPVG	HDL-C, TGs	Q5 vs. Q1 of gPVG: *β*: +0.07 (0.00, 0.14) for HDL-C; uPVG: *β* = +0.08 (0.02, 0.13) for TGs and = −0.11 (−0.18, −0.04) for HDL-C. Per 5-unit increment in uPVG: *β*: −0.02 (−0.04, −0.01) for HDL-C and +0.02 (0.01, 0.03) for TGs. Non-significant associations between the hPVG and outcomes of interest	*p* = 0.046 for gPVG; *p* = 0.003 (TGs) and *p* = 0.001 (HDL-C) for uPVG (Q5 vs. Q1)
Weston et al. [60]	Jackson Heart Study (USA)	Healthy (3,635)	M/F	51.9 to 55.5	Adherence to PDI, hPDI, uPDI (tertiles)	TC	T3 vs. T1 for PDI: +0.2; for hPDI: +0.02; for uPDI = −0.02	*p*-trend for PDI = 0.001; *p*-trend for hPDI = 0.133; *p*-trend for uPDI = 0.551

**Abbreviations:** F: females; gPVG: general pro-vegetarian diet index; HDL-C: high-density lipoprotein cholesterol; hPDI: healthful plant-based diet index; hPVG: healthful pro-vegetarian diet index; LDL-C: low-density lipoprotein cholesterol; M: males; ns: non-significant; PDI: plant-based diet index; PVG: pro-vegetarian diet index; TC: total cholesterol; TGs: triglycerides; uPDI: unhealthful plant-based diet index. Regression *β* coefficients are presented with 95% confidence intervals. All lipid measurements are given as mmol/L. Age is reported as range, mean, or range of means.

#### 3.5.4. Cross-Sectional Studies of Plant-Based Diet Indices and the Inflammatory Profile

Table 9 shows results from 3 cross-sectional studies investigating the association between the PDIs and inflammatory biomarkers. Studies featured cohorts from Sweden, Iran, and the USA, and included healthy individuals, individuals with chronic kidney disease, or individuals with overweight or obesity.

In a USA cohort of 831 healthy females, each 10-point increase in the PDI was associated with a non-significant increase in adiponectin and non-significant decreases in leptin and hsCRP [64]. However, the PDI was associated with a significant increase in IL-6. In the same study, the hPDI was associated with significantly increased concentrations of adiponectin and decreased concentrations of leptin and hsCRP. Higher adherence to the hPDI was also associated with a non-significant decrease in IL-6 concentrations. On the other hand, the uPDI was associated with significantly increased leptin concentrations, non-significantly increased hsCRP and IL-6 concentrations, and non-significantly decreased adiponectin concentrations [64]. In a Swedish cohort of males with chronic kidney disease (*n* = 418), each unit increase in the PDI was associated with significantly decreased CRP and IL-6 concentrations [61]. Likewise, in an Iranian cohort of 390 overweight or obese females, the PDI and hPDI were each associated with non-significantly decreased concentrations of hsCRP, while the uPDI was associated with the opposite [65].

#### 3.5.5. Summary of Studies Investigating Plant-Based Diet Indices and the Inflammatory Profile

Similar to the lipid profile data, greater adherence to the PDI and hPDI showed some evidence of less low-grade inflammation, whereas greater adherence to the uPDI showed increased levels of circulating inflammatory biomarkers, indicating low-grade inflammation. The small selection of studies investigating such exposures and outcomes inhibits drawing any conclusive inferences. Further research examining these associations is warranted.

**Table 9 nutrients-14-05371-t009:** Cross-Sectional Studies of Plant-Based Diet Indices and Inflammatory Biomarkers.

Reference	Country	Population (*n*)	Sex	Age (Years)	Intervention (*n*)	Outcome(s)	Results	Significance
Baden et al. [64]	USA	Healthy (831)	F	45.0	Adherence to overall PDI, hPDI, uPDI	Adiponectin (ng/mL), Leptin (ng/mL), hsCRP (mg/L), IL-6 (pg/mL)	Per 10-point increase in: PDI: adiponectin +1.1%, leptin: −1.7%, hsCRP: −7.5%, IL-6: +5.5%; hPDI: adiponectin +3.0%, leptin −7.2%, hsCRP −13.6%, IL-6 −0.7%; uPDI: adiponectin −1.6%, leptin +4.4%, hsCRP +3.3%, IL-6 +1.1%	PDI: IL-6 (*p* = 0.05); hPDI: adiponectin (*p* = 0.025), leptin (*p* < 0.001), hsCRP (*p* = 0.001); uPDI: leptin (*p* = 0.037); ns (*p* > 0.05) differences for all other outcomes
Gonzalez-Ortiz et al. [61]	Sweden	Chronic kidney disease patients (418)	M	71.0	Adherence to PDI	CRP (mg/L), IL-6 (ng/L)	Per unit increase (*β*) in PDI: CRP: −0.02 (−0.04 to −0.002), IL-6: −0.17 (- 0.33 to −0.001)	CRP (*p* = 0.03) and IL-6 (*p* = 0.04)
Pourreza et al. [65]	Iran	Overweight or obese (390)	F	18 to 48	Adherence to PDI, hPDI, uPDI	hsCRP (mg/L)	Per unit increase (*β*) in PDI: −0.01 (−0.12 to 0.10); hPDI: −0.06 (−0.15 to 0.03); uPDI: 0.07 (−0.01 to 0.17)	ns (*p* > 0.05) differences for all outcomes

**Abbreviations:** CRP: C-reactive protein; F: female; hPDI: healthful plant-based diet index; hsCRP: high-sensitivity C reactive protein; IL-6: interleukin 6; M: male; ns: non-significant; PDI: overall plant-based diet index; uPDI: unhealthful plant-based diet index. Regression *β* coefficients are presented with 95% confidence intervals. Age is reported as mean or range.

## 4. Discussion

In this review of 43 studies, individuals consuming PBDs as part of a defined intervention, or habitually, generally presented with a less pro-atherogenic lipid and lipoprotein profiles characterised by lower TC, LDL-C, and apoB concentrations, than their non-vegetarian counterparts. Similarly, intervention data on individuals consuming PBDs showed some evidence of decreased CRP/hsCRP, indicating less low-grade inflammation. Results differed by the type of PBD, generally reporting more pronounced favourable effects on lipid, lipoprotein, and inflammatory biomarkers for vegan diets, particularly non-low-fat vegan diets, compared to vegetarian diets. Notably, comparator diets also influenced results, as PBDs showed less clear benefits for the reviewed biomarkers of CVD compared to established healthy dietary patterns such as Mediterranean and ADA-recommended diets. Perhaps unsurprisingly, the most tightly controlled experiments showed the largest effect sizes, highlighting the role of adherence in mediating dietary intervention effects on CVD risk.

Here, we provide an overview of the current knowledge on plant-based dietary patterns and cardiovascular disease risk, we examine the contribution of specific plant food groups to the observed associations, including their influence on the reviewed biomarkers of CVD, and provide potential mechanistic rationales for the initiation and proliferation of atherosclerosis via lipid, lipoprotein, and inflammatory pathways.

### 4.1. Plant-Based Dietary Patterns and Cardiovascular Disease Risk

#### 4.1.1. Vegans and Vegetarian Diets and Cardiovascular Disease Risk

In large, long-term prospective cohort studies, individuals habitually consuming PBDs generally show reduced risk for CVD compared to non-vegetarians. In the Adventist Health Study II, the largest prospective study of PBDs, vegan, and lacto-ovo-vegetarian diets were associated with a non-significant reduced risk for CHD and CVD [66]. However, in a combined analysis of 5 large prospective cohort studies, including the original Adventist Health Study, vegetarians had a significant 24% lower risk of CHD compared to non-vegetarians [67]. Similarly, in the EPIC-Oxford cohort (*n* = 44,561), after a mean follow-up of 11.6 years, health-conscious lacto-ovo-vegetarians showed a 28% lower risk of CHD compared to health-conscious non-vegetarians, in the fully adjusted analysis [68]. In Taiwanese cohorts, data from the Tzu Chi Health Study (*n* = 5,050) and Tzu Chi Vegetarian Study (*n* = 8302) showed large reductions in risk for total, ischaemic, and haemorrhagic stroke for vegetarians compared to non-vegetarians [69].

#### 4.1.2. Plant-Based Indices and Cardiovascular Disease

Data from the Nurses’ Health Study (NHS) I and II and Health Professionals Follow-Up Study (HPFS) investigating associations between the PDIs and CVD outcomes observed a significant 7% and 9% lower risk of CVD mortality [70], and a 7% and 12% lower risk of CHD [71], per 10-unit increment in the PDI and hPDI, respectively, but significantly greater risk per 10-unit increment in the uPDI (8% for CVD mortality; 10% for CHD) [70,71]. Similar results have been observed for high versus low adherence to the hPDI in the same cohorts [72], and in the UK Biobank cohort, where over 150,000 individuals were followed up for a mean of 5 years, and those most adherent experienced a 17% reduced risk of CVD versus those least adherent [73]. In a meta-analysis that combined the results from Satija et al. [71] with other cohorts investigating associations between those eating PBDs or those most adherent to PBD indices and CVD outcomes, a significant 16% and 11% reduced risk for CVD and CHD was observed, respectively [1]. In a similar vein, highest versus lowest adherence to the PDI and hPDI showed significant 20% and 34% reductions in the risk of T2DM, respectively, whereas the uPDI was associated with a significant 16% increase in risk, in the NHS I and II and HPFS cohorts [56]. Results are consistent in other geographic populations, such as in one recent study in Asia [74], and a meta-analysis including cohorts from the USA, Asia, and Europe [75].

Because the PDIs measure adherence to PBDs, they do not necessitate a vegan or vegetarian diet, rather a more plant-based dietary pattern. These results therefore suggest that the cardiometabolic benefits associated with PBDs may be achieved through dietary patterns that include an abundance of healthy plant foods but are not necessarily vegan or vegetarian diets. Indeed, Shan et al. [72] compared the hPDI and other dietary scores measuring adherence to other healthy dietary patterns, including the Mediterranean diet (Alternate Mediterranean Diet Score), and dietary patterns recommended by the US Dietary Guidelines for Americans (Healthy Eating Index 2015 and Alternate Healthy Eating Index), finding similar effect sizes in the reduction of CVD risk between scores. Similarly, highest compared to lowest adherence to the PDI, hPDI, Dietary Approaches to Stop Hypertension score, Alternate Mediterranean Diet Score, and Alternate Healthy Eating Index 2010 were all associated with increased likelihood of healthy ageing, which necessitates no recent history of major chronic diseases, in a recent comparative study from the Singapore Chinese Health Study [76]. These results fit with clinical trial data on the Mediterranean diet, showing reduced risk of CVD events and mortality [77,78], and data showing a reduced risk of CVD incidence and mortality for greater adherence to the other mentioned healthy dietary patterns in meta-analyses of prospective cohort studies [1,79].

### 4.2. Plant Food Groups as Mediators of Plant-Based Dietary Pattern Associations with Cardiovascular Disease Risk

#### 4.2.1. Plant Food Group Associations with Cardiovascular Disease Risk

The relationships between PBDs and CVD risk are likely a function of the health-promoting foods that comprise such a dietary pattern, which include fruits and vegetables, whole grains, legumes, nuts, and vegetable oils. In a large meta-analysis of prospective cohort studies (*n* = 95) assessing fruits and vegetables as a combined exposure, significant 8%, 16%, and 8% reductions in the risk of CHD, stroke, and CVD were observed per 200 g/d serving, respectively [80]. Linear associations were observed for CHD and stroke mortality up to servings of 800 g/d, and similar overall associations were observed for fruits and vegetables separately [80]. Data from large Asian cohorts shows concordance, observing significant reductions in CHD, stroke, and other CVD outcomes for higher fruit and vegetable consumption, over ~7 and 11 years of follow-up [81,82].

Whole grains are similarly associated with reduced risk of cardiometabolic outcomes. In a meta-analysis of 45 prospective cohort studies, a significant 19%, 8%, and 22% reduction in the risk of CHD, stroke, and CVD, was found for 3 servings (90 g) per day of whole grain consumption, respectively, whereas refined grain consumption was not significantly associated with the aforementioned outcomes [83]. In another meta-analysis of prospective cohort studies (*n* = 16) from the same group of researchers, consumption of 3 servings per day of whole grains was associated with a significant 32% reduction in risk of T2DM, whereas refined grains showed no significant association with T2DM risk [84]. In another meta-analysis of prospective cohort studies (*n* = 14) from Harvard researchers, high versus low whole grain consumption was associated with a significant 18% reduction in CVD mortality [85].

Plant-sourced protein and food sources of plant protein such as legumes also show protective relationships with CVD outcomes. In 32-year follow-up data from the NHS and HPFS (*n* = 131,342) using repeated measures of dietary intake, plant protein was associated with a significant 12% reduction in CVD risk per 3% energy increment, after adjustment for potential confounders [86]. In a meta-analysis including 5 prospective cohort studies assessing legume consumption and cardiometabolic outcomes, a consumption of 4 weekly 100 g servings of legumes was associated with a significant 14% reduction in CHD risk, but non-significant reductions in stroke and T2DM risk [87]. Other meta-analyses of prospective cohort studies show consistency, reporting non-significant associations between legume consumption and stroke and T2DM risk [88,89]. A more recent meta-analysis of 14 prospective cohort studies also reported similar findings for CVD risk, where high versus low legume consumption was associated with a significant 10% reduction in risk of CHD and CVD [89].

Nuts are another plant food group consistently associated with reduced risk of CVD outcomes. In the same meta-analysis that observed a significant 14% reduction in CHD risk for 4 weekly 100 g servings of legumes, consumption of 4 weekly 28.4 g servings of nuts was associated with a significant 24% and 22% reduction in fatal and non-fatal CHD, respectively, a significant 13% reduction in risk of T2DM, and a non-significant 11% reduction in total stroke [87]. Larger effect sizes were observed in a more recent meta-analysis of 20 prospective cohort studies, where consumption of one serving (28 g) per day of nuts was associated with a significant 29%, 21%, and 39% reduction in the risk of CHD, CVD, and T2DM, respectively, and a non-significant 7% reduction in the risk of stroke [90].

Omega-6 fatty acids such as linolenic acid (LA) and alpha-linolenic acid (ALA), that exist in high proportions in certain nuts and vegetable oils, have also shown associations with reduced CVD risk. In a meta-analysis of 7 RCTs, LA consumption was associated with a borderline significant 12% reduction in MI, however most trials were deemed low quality [91]. In the highest quality, double-blinded trial with the longest follow-up (LA Veterans), LA-rich vegetable oil consumption showed considerable cardiovascular benefit compared to the animal-based saturated fat condition, finding a greater than 30% reduced risk of cardiovascular events and mortality [92]. Prospective cohort data shows concordance: in a meta-analysis of prospective cohort studies (*n* = 38), significant reductions in CVD mortality of 13% and 11% were observed for high versus low intakes of LA and each standard deviation increment in tissue/blood biomarkers of LA, respectively [93]. Prospective data from the NHS and HPFS cohorts also showed associations between unsaturated fatty acid intake and reductions in CVD mortality, observing a significant 27% and 10% reduced risk per 5% increment in energy (kcal) intake for polyunsaturated fatty acid (PUFA) and monounsaturated fatty acids (MUFA), respectively [94].

#### 4.2.2. Plant Food Group Effects on Lipid, Lipoprotein, and Inflammatory Biomarkers of Cardiovascular Disease

Plant-based dietary patterns may improve lipid, lipoprotein, and inflammatory profiles via the independent and additive effects that specific plant-based food constituents and groups have on such biomarkers. Soluble dietary fibre, which is abundant in certain fruits and vegetables, whole grains, and legumes, can bind bile salts in the ileum, leading to the faecal excretion of cholesterol, as well as fermenting to short-chain fatty acids, which may reduce liver cholesterol synthesis [95]. This helps to explain why meta-analyses of RCTs show significant reductions in TC (−0.12 mmol/L; 95% CI: −0.19, −0.05 mmol/L) and LDL-C (−0.09 mmol/L; 95% CI: −0.15, −0.03 mmol/L) concentrations for whole grains, and even greater reductions for the soluble-fibre rich varieties such as oats [96]. Fibre-rich foods may also lower inflammation. In a systematic review and meta-analysis of 14 RCTs, fibre or fibre-rich food interventions showed significant reductions in circulating CRP (−0.37 mg/L; 95% CI: −0.74, 0 mg/L) concentrations in overweight and obese adults [97].

Plant foods high in MUFAs and PUFAs may also mediate favourable blood lipid and lipoprotein changes. In a meta-regression of controlled feeding studies, MUFAs and PUFAs were associated with significant reductions in TC, LDL-C, HDL-C, TG, and apoB concentrations when replacing saturated fatty acids in the diet [98]. This may explain why, in a systematic review and meta-analysis of 61 RCTs assessing the lipid-altering effects of tree nut consumption, significant reductions in TC (−0.12 mmol/L; 95% CI: −0.14, −0.10 mmol/L), LDL-C (−0.12 mmol/L; 95% CI: −0.14, −0.11 mmol/L), TG (−0.02 mmol/L; 95% CI: −0.04, −0.01 mmol/L), and apoB (−3.7 mg/dL; 95% CI: −5.2, −2.3 mg/dL) concentrations were reported per 28.4 g serving/d increase in consumption, with higher intakes producing greater effects in the dose-response analysis [99]. An earlier analysis of 25 RCTs reported similar findings, where consumption of 67 g/d of nuts showed reductions in TC and LDL-C of 0.28 mmol/L and 0.26 mmol/L, respectively [100]. Nuts are also rich in plant sterols, which may explain why they lower lipid concentrations further than would be expected based solely on their fatty acid profile [101]. In a RCT involving 42 young male students, swapping 30 g of butter for equivalent servings of a PUFA-rich margarine or a PUFA- and plant-sterol-rich margarine showed significant reductions in LDL-C concentrations of 6% and 11%, respectively [102].

Nuts may also reduce low-grade inflammation. Interventions administering almonds and walnuts have shown reductions in CRP concentrations in some but not all trials [103], and the PREDIMED study observed a reduction in IL-6 levels for the nut intervention group compared to the control group, however no significant differences were observed in CRP concentrations [77]. Another study that administered high-PUFA diets fortified with walnuts and walnut oil showed a 75% and ~45% reduction in CRP concentrations in the ALA and LA conditions, respectively [104]. In addition, the ALA group showed significant reductions in IL-6 and TNF-α concentrations compared to the LA group. However, in a systematic review and meta-analysis of 22 RCTs, nut consumption showed no significant changes in most inflammatory biomarkers, but was associated with significantly reduced leptin (−0.71 mg/dL; 95% CI: −1.11, −0.30 mg/dL) concentrations [105].

Food sources of plant protein may also mediate favourable lipid and lipoprotein biomarker profiles. In a systematic review and meta-analysis of 112 RCTs, the substitution of plant protein for animal protein showed reductions in LDL-C (−0.16 mmol/L; 95% CI: −0.20, −0.12 mmol/L), non-HDL-C (−0.18 mmol/L; 95% CI: −0.22, −0.14 mmol/L), and apoB (−5 mg/dL; 95% CI: −6, −3 mg/dL) concentrations [106]. A systematic review and meta-analysis of 43 RCTs investigating the effect of soy protein on lipid concentrations demonstrated significant reductions in TC (−0.17 mmol/L; 95% CI: −0.24, −0.09) and LDL-C (−0.12 mmol/L; 95% CI: −0.17, −0.07 mmol/L) concentrations per 25 g/d serving compared to non-soy protein controls [107]. Similarly, there are data suggesting benefits for low-grade inflammation for certain food sources of plant protein. For example, non-soy legumes showed borderline significant reductions in CRP and hsCRP (−0.21 mg/L; 95% CI: −0.44, 0.02) concentrations in a meta-analysis of 8 trials, and significant reductions in sensitivity analyses [108].

The Portfolio Diet, a dietary pattern combining the aforementioned food groups, offers insight into the potential effect of a combination of lipid, lipoprotein, and inflammatory modulating food groups on such outcomes. In a systematic review and meta-analysis of 7 RCTs, large (significant) reductions in TC (−0.81 mmol/L; 95% CI: −0.98, −0.64 mmol/L), non-HDL-C (−0.83 mmol/L; 95% CI: −1.03, −0.64 mmol/L), LDL-C (−0.73 mmol/L; 95% CI: −0.89, −0.56 mmol/L), TG (−0.28 mmol/L; 95% CI: −0.42, −0.14 mmol/L), apoB (−19 mg/dL; 95% CI: −23, −15 mg/dL), and CRP (−0.53 mg/L; 95% CI: −1.01, −0.15 mg/L) concentrations were observed for Portfolio Diet groups compared to control diet groups [109]. Indeed, our review included a RCT testing the Portfolio Diet versus 20 mg of lovastatin therapy, finding non-inferiority for lipid-lowering purposes [38].

### 4.3. Mechanisms Underpinning Plant-Based Dietary Pattern Associations with Reduced Cardiovascular Risk

#### 4.3.1. Lipid and Lipoprotein Profiles and Atherosclerotic Cardiovascular Disease

Associations between PBDs and reduced risk of CVD outcomes are likely partially mediated by favourable changes in lipid and lipoprotein profiles. As mentioned in the introduction, LDL-C concentrations are the target of conventional CVD risk-reduction therapy because when lowered, risk of CHD is lowered in a dose-dependent, log-linear fashion, across multiple lines of evidence [12]. Contemporary evidence has demonstrated that the associated reduction in LDL particle concentration, and more precisely, apoB-containing lipoprotein particle concentration, is driving the reduction in CVD risk. The overwhelming evidence relating LDL and apoB-containing lipoprotein particles to atherogenesis allows for causal inference [12]. This is because all apoB-containing lipoprotein particles, of which ~90% are LDL, can enter the arterial intima and cause atherosclerosis [12,13]. In addition, while LDL-C concentrations often mirror apoB concentrations, depending on the methodology of determining discordance, between 20% and 60% of individuals can be classed as discordant, and apoB concentrations will better explain CVD risk in these individuals [110,111,112,113,114].

The “response-to-retention” model of atherogenesis, which states that the key initiating event of atherogenesis is the retention of cholesterol-rich apoB-containing lipoprotein particles within the arterial wall [115], helps to explain the causal association between apoB-containing lipoprotein particle concentrations and atherosclerosis. As all apoB-containing lipoprotein particles less than 70 nm in diameter (chylomicron remnants, VLDL, intermediate-density lipoprotein, LDL, and lipoprotein (a)) can enter the subendothelial space in the arterial wall, they can all be retained [116]. While there are many factors that can affect whether apoB particles are retained in the arterial intima, e.g., the ability for apoB particles to interact and bind with arterial wall proteoglycans, the residence time of LDL and apoB particles in plasma is believed to be the critical factor in atherogenesis, as it determines the exposure of the arterial tissue to atherogenic particles, increasing the likelihood that the particles will undergo proatherogenic intravascular modifications [110,116]. Once retained, apoB-containing lipoprotein particles trigger a cascade of pro-atherogenic events that initiate and propagate atheroma development, including oxidation, the recruitment of monocytes into the arterial wall, and their transformation to macrophages. Ultimately, the retention and modification of apoB-containing lipoprotein particles evokes innate and adaptive immune responses that drive inflammation into the artery wall, forming foam cells and eventually atherosclerotic plaque [116]. Further support for this hypothesis stems from the observation that when the rate of retention of apoB-containing lipoprotein particles is reduced, atherogenesis is also reduced, despite no changes to the influx of particles within the arterial intima (in animal models) [115].

Other lipid and lipoprotein particle concentrations may also play a role in the progression of atherosclerotic CVD. Triglyceride concentrations are associated with CVD independently of LDL-C and non-HDL-C concentrations [117,118]. When adjusting for apoB concentrations, however, associations may attenuate: in a combined analysis of the UK Biobank, FOURIER, and IMPROVE-IT trials (*n* = 389,529), while non-HDL-C, TG, and apoB concentrations each associated with MI, only apoB concentrations associated with MI when assessed together [119]. Similarly, in Mendelian randomisation analysis (*n* = 654,783), associations between LDL-C and TG concentrations were nullified after adjustment for apoB-containing lipoprotein particles [120]. Triglyceride-rich lipoprotein particles such as VLDL, which contain an apoB molecule, may thus mediate the association between TG concentrations and CVD. The cholesterol content of the VLDL particles may also contribute to the atherogenicity of apoB-containing lipoprotein particles: in a recent analysis of the Copenhagen General Population Study (*n* = 25,480), VLDL-C concentrations explained approximately half of the risk of MI conferred by apoB-containing lipoprotein particle concentrations, whereas VLDL TG concentrations did not [121]. Despite this, LDL particle concentrations drive the majority of atherogenic lipoprotein risk because they represent ~90% of all circulating apoB particles in most individuals [110].

Differences in lipoprotein particle subclass profiles may also mediate atherogenesis. As previously mentioned, a pro-atherogenic phenotype is characterised by a preponderance of small, dense LDL [14,15,16], small HDL [16,17], and large VLDL particle concentrations [16]. Indeed, previous work from our group reported more small LDL, large VLDL, and less large HDL particles in metabolically unhealthy individuals, with and without obesity [122], a finding that has also been associated with insulin resistance [123]. Small, dense LDL particles may confer increased risk of atherosclerotic CVD due to longer residence time in plasma and enhanced arterial wall penetration ability, which combine to result in greater potential to undergo pro-atherogenic intravascular modification [116]. Accordingly, small, dense LDL particles have been associated with CVD independent of LDL-C concentrations in multiple prospective cohort studies [124,125,126], and in a randomised trial [127], and predicted the rate of CHD independently of LDL-C, TG, HDL-C, and apoB concentrations in the Québec Cardiovascular Study [128]. Other components of the pro-atherogenic lipoprotein subclass profile have similarly been associated with unfavourable CVD outcomes in a cross-sectional study, where men with higher than median concentrations of either small HDL or large VLDL particles were more likely to have extensive CHD than those with lower than median concentrations [129].

Characterising independent associations between these subclasses and CVD is difficult because they cluster with other components of metabolic syndrome [130]. For example, those with a preponderance of small, dense LDL particles tend to also have greater total LDL particle concentrations [131], which may drive the increased risk of atherosclerosis in individuals with that phenotype. Data from the Multi-Ethnic Study of Atherosclerosis tested this idea, finding the association between small, dense LDL particles and carotid intima-media thickness to attenuate after adjustment for total LDL particle concentration [132]. Supporting this finding, a systematic review found that for most studies, the relationship between LDL particle size and cardiovascular outcomes is attenuated after adjustment for total LDL particle concentration [133], and a recent prospective analysis of the UK Biobank cohort (*n* = 96,126) reported that despite significant associations between VLDL and LDL particle size and MI in crude analyses, only apoB and lipoprotein(a) particle concentrations were significantly associated with MI in the fully adjusted analysis [134].

Finally, HDL-C concentration has been traditionally thought of as important in the reduction of CVD risk, because of inverse associations with CVD risk in prospective cohort studies [135]. For example, the large Framingham cohort study observed significant decreases in CVD risk of 2–3% per 0.26 mmol/L increase in HDL-C concentration [136]. Despite these associations, randomised investigations that increased HDL-C concentration pharmacologically have failed to show reductions in CVD risk [135,137,138], questioning this hypothesis. The mechanism underpinning the anti-atherogenic effects of HDL-C concentration relates to reverse-cholesterol transport, where HDL particles take cholesterol from peripheral tissues, such as cholesterol-rich macrophages, back to the liver [135]. Other metrics such as cholesterol efflux capacity and total HDL particle concentration may be more accurate markers of CVD risk [139,140], but their utility may be secondary to more established markers, as associations between total HDL particle concentration and HDL particle size and MI were attenuated after adjustment for apoB and lipoprotein(a) particle concentrations in a recent analysis of the UK Biobank cohort [134].

In summary, the associations between PBDs and decreased TC, LDL-C, and apoB concentrations represent reduced risk for atherosclerotic CVD for those consuming PBDs, and the decrease in HDL-C concentration often observed on such diets is unlikely to alter this conclusion.

#### 4.3.2. Inflammation and Atherosclerotic Cardiovascular Disease

As low-grade inflammation plays a role in all aspects of atherosclerotic CVD [141], protective associations between PBDs and CVD risk may potentially reflect favourable changes in inflammatory biomarkers. Despite the significant reductions in CVD risk observed with aggressive lowering of LDL-C concentration, patients with established CVD still experience residual risk [142], which may, at least in part, be due to a range of risk factors, including inflammation [143]. Evidence from large lipid-lowering trials has allowed for the emergence of a “dual targets” approach to CVD risk reduction, that is, the achievement of low LDL-C (< 1.81 mmol/L) and hsCRP (<2 mg/dL) concentrations [143]. In the 2.5-year-long PROVE-IT-TIMI 22 trial (*n* = 3,745), while individuals in the highest versus lowest quartiles of LDL-C concentration exhibited a significant 70% increased risk of coronary events in the fully adjusted analysis (including adjustment for hsCRP concentration), associations with highest versus lowest quintiles of hsCRP concentrations showed an equal increase in risk in a similar fully adjusted model (adjusting for LDL-C concentration) [144]. In addition, the larger IMPROVE-IT trial (*n* = 18,144) reported a significant 27% reduction in CVD events after a median follow-up of 74 months for those reaching this “dual target” of low LDL-C and CRP concentrations compared to those reaching neither [145]. Large prospective cohort studies have thus used CRP concentration as a predictor of CVD risk, demonstrating significant independent associations with future CVD risk [22,146]. Other trials have reported similar associations between other markers of low-grade inflammation such as interleukin-1 beta (IL-1β) and IL-6 concentrations and heightened CVD risk [143], strengthening the certainty of evidence. Further, evidence from trials administering inflammation-lowering therapeutic agents have suggested an independent causal role of inflammation in atherosclerotic CVD. The CANTOS (*n* = 10,061) [147] and COLCOT (*n* = 4745) [148] trials both administered anti-inflammatory therapeutic agents in high-risk individuals, reporting a significant 15% reduction in cardiovascular death and 23% reduction in CVD outcomes after a median of 24- and ~22-month-follow-up, respectively.

The mechanisms that may mediate these reductions in hard CVD outcomes likely relate to the role of inflammation in atherosclerosis. When apoB-containing lipoproteins enter the arterial intima and deposit cholesterol, this accumulation of cholesterol ignites a subintimal inflammatory response [143]. In addition, inflammation-mediated endothelial dysfunction leads to the upregulation of adhesion molecules that promote the infiltration of inflammatory cells such as monocytes and leukocytes to early plaque sites, where the monocytes can become pro-inflammatory macrophages [143,149]. The resultant inflammasome releases upstream inflammatory cytokines such as IL-1β that activate inflammatory cells and produce IL-6, which stimulates the production of CRP, amplifying the inflammatory cascade and promoting atherosclerosis [143,150]. Other inflammatory cells involved in this cascade promote the production of cytokines such as TNF-α [143], which may promote endothelial dysfunction and enhanced LDL particle transcytosis into the arterial intima, also enhancing LDL particle retention [151]. Alongside its role in atherogenesis, inflammation may destabilise fibrous atherosclerotic plaques through the action of cytokines such as IL-1β that degrade collagen in the extracellular matrix formed within the plaque [143,152]. This ultimately increases the risk for rupture, which has been demonstrated to be the suspected cause of death in ~60–75% of patients with sudden coronary death and thrombosis [153,154].

While atherosclerosis is dependent on the infiltration of apoB-containing lipoproteins to the arterial intima, inflammation may play an independent and causal role in the initiation and proliferation of all stages of atherosclerotic plaque development, highlighting the interplay between the reviewed lipid, lipoprotein, and inflammatory biomarkers of CVD. The decreases in markers of low-grade inflammation often observed in PBD interventions may thus partly mediate the reduced CVD risk observed for habitual PBD consumers.

### 4.4. Limitations and Future Directions

The literature has traditionally focused on lipid profiles, with less investigation into lipoprotein and inflammatory profiles, which may be more sensitive markers of CVD risk. In addition, the studies investigating lipoprotein and inflammatory profiles largely focused on a limited number of such markers, namely, apoB and CRP/hsCRP, respectively. Further investigation into lipoprotein subclass profiles and other inflammatory biomarkers of CVD may shed further light on important relationships between PBDs and CVD risk.

Another limitation is the type of comparator diet in trials. A lot of trials compared a PBD to a usual-diet or Western-style-diet comparator, often reporting favourable results for the PBD compared to such controls. Indeed, when PBDs were compared to more established healthy dietary patterns, e.g., Mediterranean diet, results were less profound for the markers of interest. To better understand the utility of a PBD in CVD risk reduction, future trials should continue to compare PBDs to established healthy dietary patterns to further elucidate benefits inherent to PBDs for lipid, lipoprotein, and inflammatory biomarkers of CVD.

Finally, more research is warranted using PBD scores or indices, such as the PDIs, and investigating their relation with lipids, lipoprotein, and inflammatory biomarkers of CVD. These scores measure adherence to a more plant-based dietary pattern, and not a fully PBD, and therefore offer utility in assessing how more gradual shifts towards plant-based dietary patterns may affect such biomarkers of CVD risk. This exposure is of importance as populations are encouraged to shift towards more plant-based dietary patterns as part of climate change mitigation strategies. Further, data employing this exposure should ideally follow cohorts prospectively and with repeated measures of diet, to allow for assessment of temporality and how trends in dietary intake may affect lipid, lipoprotein, and inflammatory biomarkers of CVD.

## 5. Conclusions

In summary, the available RCT and prospective cohort evidence shows favourable relationships between PBDs and most lipid and lipoprotein biomarkers, and some inflammatory biomarkers. Because pro-atherogenic lipid, lipoprotein, and inflammatory profiles can cause and accelerate atherosclerosis, the observed relationships may add context to the reduced CVD risk observed in those consuming defined PBDs, or those consuming more plant-based dietary patterns. Because the research literature has largely focused on the relation between PBDs and traditional plasma lipid profiles and CRP/hsCRP, with a relative lack of research on other inflammatory biomarkers and lipoprotein subclass profiles, future research should investigate these outcomes to further elucidate PBD associations with biomarkers of CVD.

## Figures and Tables

**Figure 1 nutrients-14-05371-f001:**
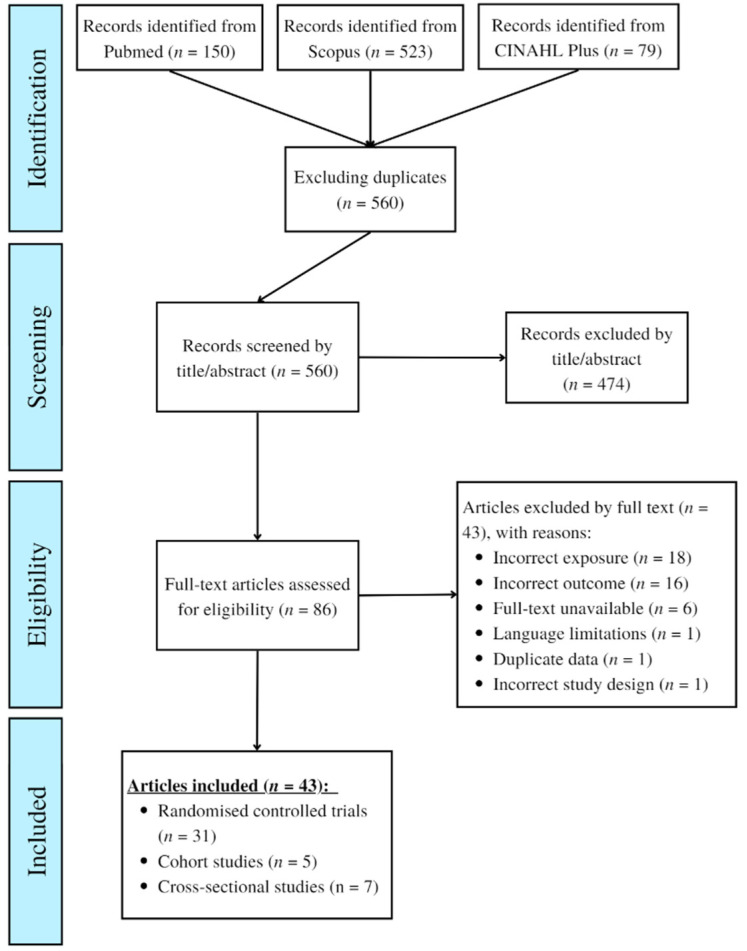
Flow Chart of Database Search and Article Eligibility.

**Table 1 nutrients-14-05371-t001:** Eligibility Criteria.

PICOS	Inclusion Criteria	Exclusion Criteria
Population	Adults (>18 years old)	Children and adolescents (<18 years old)
Intervention/ Exposure	Plant-based diets (i.e., vegan/vegetarian diets) administered as an intervention, or plant-based dietary scores measured by food frequency questionnaire	Non-plant-based dietary patterns; unclear definitions or measurement of dietary exposures.
Comparison	For RCTs: control groups consisting of usual diet or another dietary pattern. For cohort studies: plant-based diet groups compared to non-vegetarian diet groups. For diet scores/indices, comparisons between quintiles or continuous measures (e.g., per 10-unit increment) were included	Comparisons where other exposures are included with diet vs. comparator, e.g., exercise, fasting, etc.; multifactorial interventions.
Outcomes	Lipid biomarkers: TC, LDL-C (directly measured LDL-C was included over calculated LDL-C where both were reported); HDL-C; Non-HDL-C; VLDL-C; TGs Lipoprotein biomarkers: VLDL-P concentrations; IDL-P concentrations; LDL-P concentrations; HDL-P concentrations; Non-HDL-P concentrations; apoB; Small, medium, large VLDL-P, LDL-P, HDL-P concentrations; Mean VLDL-P size; Mean LDL-P size; Mean HDL-P size Inflammatory biomarkers: C3; hsCRP/CRP; IL-6; TNF-α; Adiponectin; Leptin; LAR; Resistin; PAI-1; WBCs; Neutrophils; Lymphocytes; NLR; Monocytes; Basophils; Eosinophils; GlycA	All other outcomes Postprandial lipid outcomes Self-reported outcomes without objective measurements
Study type	RCTs (parallel, crossover, metabolic ward), prospective cohort studies, cross-sectional studies (only for studies using dietary scores/indices), systematic reviews and meta-analyses (of randomised controlled trials and/or prospective cohort studies)	Non-randomised trials, intervention trials without a control/comparator, reviews, case reports/series, editorials, commentaries, meeting abstracts, studies with legitimate expressions of concern

**Abbreviations:** apoB: apolipoprotein B; C3: complement component 3; CRP: C-reactive protein; GlycA: glycoprotein A; HDL-C: high-density lipoprotein cholesterol; HDL-P: high-density lipoprotein particle; hsCRP: high-sensitivity C reactive protein; IDL-P: intermediate-density lipoprotein particle; IL-6: interleukin 6; LAR: leptin-to-adiponectin ratio; LDL-C: low-density lipoprotein cholesterol; LDL-P: low-density-lipoprotein particle; NLR: neutrophil–lymphocyte ratio; PAI-1: plasminogen activator inhibitor 1; RCTs: randomised controlled trials; TC: total cholesterol; TGs: triglycerides; TNF-α: tumour necrosis factor alpha; VLDL-C: very-low-density lipoprotein cholesterol; VLDL-P: very-low-density lipoprotein particle; WBCs: white blood cells.

**Table 2 nutrients-14-05371-t002:** Literature Search Strategy.

Search	Search Terms
#1: Population	Human adults aged 19+ (filter)
#2: Intervention/ Exposure	(“plant-based diet index” OR “plant-based diet” OR “plant-based dietary pattern” OR “plant-based diets” OR “plant-based diet scores” OR “plant-based dietary scores” OR “vegan” OR “vegan diet” OR “vegetarian” OR “vegetarian diet”)
#3: Study Types	(“cohort study” OR “follow-up” OR “randomized controlled trial” OR “randomised controlled trial” OR “RCT” OR “clinical trial” OR “meta-analysis” OR “cross-sectional” OR “case-control”)
#4: Outcomes	(“Lipids” OR “Plasma lipids” OR “Cholesterol” OR “Lipoproteins” OR “Subclasses” OR “Profiles” OR “Low-density lipoprotein” OR “LDL” OR “High-density lipoprotein” OR “HDL” OR “Triglycerides” OR “non-high-density lipoprotein” OR “non-HDL” OR “Small LDL” OR “Large LDL” OR “Intermediate-density lipoprotein” OR “IDL” OR “Very-low-density lipoprotein” OR “VLDL” OR “Apolipoprotein B” OR “apoB” OR “Inflammation” OR “inflammatory biomarkers” OR “complement component 3” OR “C3” OR “acute-phase response proteins” OR “APRPs” OR “high-sensitivity C-reactive protein” OR “hsCRP” OR “CRP” OR “C-reactive protein” OR “C reactive protein” OR “pro-inflammatory cytokines” OR “pro inflammatory cytokines” OR “interleukin 6” OR “IL6” OR “IL-6” OR “TNF-α” OR “TNFa” OR “TNF-alpha” OR “tumour necrosis factor alpha” OR “tumor necrosis factor alpha” OR “adipocytokines” OR “adiponectin” OR “leptin” OR “leptin-adiponectin ratio” OR “resistin” OR “PAI-1” OR “plasminogen activator inhibitor 1” OR “white blood cells” OR “white blood cell count” OR “leukocytes” OR “neutrophils” OR “lymphocytes” OR “neutrophil-lymphocyte ratio” OR “monocytes” OR “basophils” OR “eosinophils” OR “glycoprotein A” OR “GlycA”)
#5: Additional Filters	Language: English | PubMed: Title/Abstract | Scopus: Title/Abstract/Keywords
#6	#1 AND #2 AND #3 AND #4 AND #5

**Abbreviations:** apoB: apolipoprotein B; APRPs: acute-phase response proteins; C3: complement component 3; CRP: C-reactive protein; GlycA: glycoprotein A; HDL: high-density lipoprotein particle; hsCRP: high-sensitivity C reactive protein; IDL: intermediate-density lipoprotein; IL-6: interleukin 6; LDL: low-density lipoprotein; PAI-1: plasminogen activator inhibitor 1; RCTs: randomised controlled trials; TNF-α: tumour necrosis factor alpha; VLDL: very-low-density lipoprotein.

**Table 4 nutrients-14-05371-t004:** Prospective Cohort Studies of Plant-Based Diets and Lipid Profiles.

Reference	Cohort (Country)	Population (*n*)	Sex	Age (Years)	Exposure (*n*)	Follow-up	Outcomes	Results
Chiu et al. [51]	MJ Health Screening Database (Taiwan)	Healthy (5,734)	M/F	48.9 for composite VD and NVD	LOV (624) vs. LV (173) vs. VG (159) vs. NVD (4778)	Median 2.12 years	High TC (≥ 5.17), high LDL-C (≥ 3.36), low HDL-C (< 1.03 [M]/1.29 [F]), high TGs (≥ 1.70)	Fully adjusted ORs (with 95% CIs): LOV vs. NVD: high TC: 0.99 (0.94, 1.03); high LDL-C: 1.01 (0.95, 1.06); low HDL-C: 1.08 (1.03, 1.12); high TGs: 1.04 (0.99, 1.09); VG vs. NVD: high TC: 0.97 (0.89, 1.06); high LDL-C: 0.92 (0.82, 1.04); low HDL-C: 1.04 (0.95, 1.14); high TGs: 1.00 (0.91, 1.10); LV vs. NVD: high TC: 1.05 (0.98, 1.13); high LDL-C: 1.01 (0.93, 1.10); low HDL-C: 0.99 (0.90, 1.10); high TGs: 1.02 (0.95, 1.11). *p*-values not reported.
Shang et al. [50]	MJ Health Screening Database (Taiwan)	Healthy (93,209)	M/F	NVD: 36.8; PV: 43.5; LOV: 37.9; VG: 44.1	NVD (85,319) vs. PV (2461) vs. LOV (4313) vs. VG (1116)	Mean 3.75 years	Low HDL-C (< 1.03 [M]/1.29 [F]), high TGs (≥ 1.69)	Fully adjusted HRs (with 95% CIs): NVD vs. VG: low HDL-C: 0.72 (0.62, 0.84); high TGs: 0.86 (0.74, 1.09); PV vs. VG: low HDL-C: 0.70 (0.57, 0.84); high TGs: 0.85(0.71, 1.02); LOV vs. VG: low HDL-C: 0.98 (0.83, 1.17); high TGs: 0.92 (0.78, 1.09). *p*-values not reported.

**Abbreviations:** F: female; HDL-C: high-density lipoprotein cholesterol; LDL-C: low-density lipoprotein cholesterol; LOV: lacto-ovo-vegetarian diet; LV: lacto-vegetarian diet; M: male; NVD: non-vegetarian diet; PV: pesco-vegetarian diet; TC: total cholesterol; TGs: triglycerides; VD: vegetarian diet; VG: vegan diet. All lipid measurements are given as mmol/L. Age is reported as mean.

**Table 5 nutrients-14-05371-t005:** Randomised Controlled Trials of Plant-Based Diets and Lipoprotein Profiles.

Reference	Country	Population (*n*)	Sex	Age (Years)	Intervention	Study Length/ Design	Outcomes	* Results	Significance
Cooper et al. [44]	USA	Healthy (15)	M/F	All: 28.0	LOV vs. typical USA diet	6 weeks: 3 weeks × 2 (crossover)	apoB	−7.6	*p* < 0.05
Djekic et al. [46]	Sweden	Overweight (31)	M/F	LOV: 67.0; NVD: 68.0	Isocaloric LOV (16) vs. NVD (15) [both adhering to Nordic Recommendations]	12 weeks: 4 weeks × 2 (crossover) with a 4-week washout in between	apoB	−2.1	ns (*p* > 0.05)
Hall et al. [37]	USA	Overweight (20)	M/F	All: 29.9	LFPBD vs. ABKD	4 weeks: 2 weeks × 2 (crossover)	VLDL-P, LDL-P, HDL-P, VLDL-P size, LDL-P size, HDL-P size, large LDL-P, medium LDL-P, small LDL-P, large HDL-P, medium HDL-P, small HDL-P, apoB	VLDL-P: +27.9; LDL-P: −443.0; HDL-P: −3.5; large LDL-P: +122.0; medium LDL-P: −176.0; small LDL-P: −438.0; large HDL-P: −1.0; medium HDL-P: +0.8; small HDL-P: −3.8; VLDL-P size: +1.6; LDL-P size: +0.1; HDL-P size: 0.0; apoB: −19.6	*p* < 0.001 for all except large LDL-P (*p* = 0.002), medium LDL-P (*p* = 0.013), medium HDL-P (*p* = 0.05) and VLDL-P, LDL-P, and HDL-P size (all ns, or *p* > 0.05)
Hunt et al. [45]	USA	Healthy (21)	F	All: 33.2	LOV vs. NVD	8 weeks: 4 weeks × 2 (crossover)	apoB	−6	*p* = 0.05
Jenkins et al. [38]	Canada	Hyperlipidaemic (34)	M/F	All: 58.4	Statin vs. Portfolio Diet vs. low-saturated-fat control diet	3 × 1 month (crossover) intervention periods with a 2-to-6-week washout period between	apoB	−26	*p* < 0.005; result for the Portfolio Diet vs. statin group was non-significantly different
Jenkins et al. [39]	Canada	Overweight and hyperlipidaemia (44)	M/F	LCPBD: 56.1; LFLOV: 57.8	LCPBD (22) vs. LFLOV (22)	1-month parallel, metabolically controlled study	apoB	LCPBD: −31; LFLOV diet: −19	LCPBD had significantly lower apoB (*p* = 0.001) vs. LFLOV diet
Jenkins et al. [40]	Canada	Overweight and hyperlipidaemia (39)	M/F	LCPBD: 57.6; LFLOV: 55.3	LCPBD (20) vs. LFLOV (19)	6 months (parallel)	apoB	LCPBD: −22; LFLOV: −15	LCPBD had significantly lower apoB (*p* < 0.001) vs. LFLOV diet
Ling et al. [36]	Finland	Healthy (18)	M/F	VG: 48.0; NVD: 37.5	Uncooked VG (including fermented foods) vs. mixed NVD	4 weeks (parallel)	apoB	−21	No significance tests were conducted between groups. The VG diet significantly lowered apoB (*p* < 0.01) vs. baseline values
Shah et al. [33]	USA	CHD (100)	M/F	VG: 63.0; AHA: 59.5	VG (50) vs. AHA-recommended diet (50)	8 weeks (parallel)	LDL-P, HDL-P, large VLDL-P; small LDL-P; large HDL-P; VLDL-P size, LDL-P size; HDL-P size	LDL-P: −2; HDL-P: +3; large VLDL-P: 0; small LDL-P: +29; large HDL-P: −0.7; VLDL-P size: +1; LDL-P size: −0.1; HDL-P size: −0.1	ns (*p* > 0.0015) differences between groups for all outcomes using linear regression analysis (Bonferroni correction applied)

**Abbreviations:** ABKD: animal-based ketogenic diet; AHA: American Heart Association; apoB: apolipoprotein B; CHD: coronary heart disease; F: females; HDL-P: high-density lipoprotein particle concentrations; LCPBD: low-carbohydrate plant-based diet; LDL-P: low-density lipoprotein particle concentration; LFLOV: low-fat lacto-ovo-vegetarian diet; LFPBD: low-fat plant-based diet; LOV: lacto-ovo-vegetarian diet; M: males; ns: non-significant; NVD: non-vegetarian diet; PBD: plant-based diet; VG: vegan; VLDL-P: very-low-density lipoprotein particle concentrations. *Results are presented as the difference between interventions (PBD vs. comparison) except for Jenkins et al. [39,40]. Lipoprotein particle concentrations were measured as nmol/L (LDL-P, VLDL-P) or μmol/L (HDL-P). Lipoprotein particle size was measured as nm for all. apoB was measured as mg/dL. Age is reported as mean or median.

**Table 7 nutrients-14-05371-t007:** Prospective Cohort Studies of Plant-Based Diet Indices and Lipid Profiles.

Reference	Cohort (Country)	Population (*n*)	Sex	Age (Years)	Exposure (*n*)	Follow-up	Outcomes	Results
Kim et al. [57]	KoGES (Korea)	Healthy (5,646)	M/F	49.0 to 52.4	Adherence to PDI, hPDI, uPDI	Median of 8 years	Low HDL-C (<1.03 [M]/1.29 [F]), hypertriglyceridaemia (TGs > 1.70)	ns (*p* > 0.05) associations between PDI and hPDI and outcomes; HRs for Q5 vs. Q1 for uPDI: 1.25 (95% CI: 1.09, 1.43) for low HDL-C; 1.26 (95% CI: 1.08, 1.46) for hypertriglyceridaemia
Lee et al. [58]	KoGES (Korea)	Healthy (16,068)	M/F	49.9 to 53.7	Adherence to PDI, hPDI, uPDI	14 years	Dyslipidaemia (One of the following: TGs ≥ 5.18; TC ≥ 6.12; HDL-C < 1.00; LDL-C ≥ 4.10; or use of any anti-dyslipidaemia medication)	Multivariable-adjusted HRs for highest vs. lowest quintiles for dyslipidaemia were 0.78 (95% CI: 0.69–0.88) for PDI, 0.63 (95% CI: 0.56–0.70) for hPDI, and 1.48 (95% CI: 1.30–1.69) for uPDI (*p*-trend < 0.001 for all)
Zhu et al. [59]	8 European countries	Overweight/obese with pre-diabetes (710)	M/F	57.0	Novel plant-based diet score	3 years	LDL-C	Fully adjusted result for longitudinal association with yearly changes in LDL-C: −0.03 (95% CI: −0.07, 0.001); ns (*p* = 0.057)

**Abbreviations:** F: female; HDL-C: high-density lipoprotein cholesterol; hPDI: healthful plant-based diet index; KoGES: Korean Genome and Epidemiology Study; LDL-C: low-density lipoprotein cholesterol; M: male; ns: non-significant; PDI: overall plant-based diet index; TC: total cholesterol; TGs: triglycerides; uPDI: unhealthful plant-based diet index. All lipid measurements are given as mmol/L. Age is reported as either the range of means, or median.

## Data Availability

Not applicable.

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
