# Peer review of "Plant-Based Diets and Lipid, Lipoprotein, and Inflammatory Biomarkers of Cardiovascular Disease: A Review of Observational and Interventional Studies"

_nutrients, 2022, doi:10.3390/nu14245371_

Round 1

Reviewer 1 Report

I read the current manuscript with great interest.

IMHO this is a very well written and comprehensive review, including all the necessary literature. 

I have two suggestions:

1) It would be really nice if authors could add a graphical abstract describing the relationships between PBDs 71 and lipid, lipoprotein, and inflammatory biomarkers of CVD.

2) In discussion section, authors should add a subchapter describing the main limitations of included literature and what should be implemented.

Author Response

Dear Reviewer 1,

We thank you for your review and comments. We agree with your points, and have addressed both of them by adding a graphical abstract to our submission, and by including a "Limitations and Future Directions" subheading and section to the Discussion.

Kind regards,

Patrick, on behalf of the co-authors

Reviewer 2 Report

The available RCT and prospective cohort evidence shows favourable  relationships between PBDs and most lipid and lipoprotein biomarkers, and some inflammatory biomarkers.

Because pro-atherogenic lipid, lipoprotein, and inflammatory profiles can cause and accelerate atherosclerosis, the observed relationships may add context to  the reduced CVD risk observed in those consuming defined PBDs, or those consuming  more plant-based dietary patterns.

The research literature has largely focused on the relation between PBDs and traditional plasma lipid profiles and CRP/hsCRP, with a relative lack of research on other inflammatory biomarkers and lipoprotein subclass profiles, future research should investigate these outcomes to further elucidate PBD associations with biomarkers of CVD.

Similarly, more research investigating PDI associations with biomarkers of CVD is warranted to better understand the effects of adhering to a  more plant-based dietary pattern, and not a fully PBD, on intermediate biomarkers of  CVD.  

Author Response

Dear Reviewer 2,

On behalf of myself and my co-authors, thank you very much for your review. In your Comments and Suggestions, we don't see any particular comments or suggestions, just our conclusions from the paper.

Would you be able to provide us with comments and/or suggestions, if applicable?

We look forward to hearing back from you.

Regards,

Patrick, on behalf of the co-authors